# CRYSTALBOX: EFFICIENT MODEL-AGNOSTIC EXPLANATIONS FOR DEEP RL CONTROLLERS

## ABSTRACT

Practical adoption of Reinforcement Learning (RL) controllers is hindered by a lack of explainability. Particularly, in input-driven environments such as computer systems where the state dynamics are affected by external processes, explainability can serve as a key towards increased real-world deployment of RL controllers. In this work, we propose a novel framework, CrystalBox, for generating black-box post-hoc explanations for RL controllers in input-driven environments. CrystalBox is built on the principle of separation between policy learning and explanation computation. As the explanations are generated completely outside the training loop, CrystalBox is generalizable to a large family of input-driven RL controllers. To generate explanations, CrystalBox combines the natural decomposability of reward functions in systems environments with the explanatory power of decomposed returns. CrystalBox predicts these decomposed future returns using on-policy Q-function approximations. Our design leverages two complementary approaches for this computation: sampling- and learning-based methods. We evaluate CrystalBox with RL controllers in real-world settings and demonstrate that it generates high-fidelity explanations.

## 1 INTRODUCTION

Deep Reinforcement Learning (DRL) based solutions outperform manually designed heuristics in many computer systems and networking problems in lab settings. DRL agents have been successful in a wide variety of areas, such as Adaptive Bitrate Streaming (Mao et al., 2017), congestion control (Jay et al., 2019), cluster scheduling (Mao et al., 2019b), and network traffic optimization (Chen et al., 2018). However, because DRL agents choose their actions in a black-box manner, systems operators are reluctant to deploy them in real-world systems (Meng et al., 2020). Hence, similar to many ML algorithms, the lack of explainability and interpretability of RL agents has triggered a quest for eXplainable Reinforcement Learning algorithms and techniques (XRL).

There are two major research directions in explainability of deep RL. The first line of work, which can be described as feature-based methods, transfer established XAI results developed for supervised learning algorithms to deep RL settings. They focus on tailoring commonly used post-hoc explainers for classification and regression tasks, such as saliency maps (Zahavy et al., 2016; Iyer et al., 2018; Greydanus et al., 2018; Puri et al., 2019) or model distillation (Bastani et al., 2018; Verma et al., 2018; Zhang et al., 2020). While such adapted techniques work well for some RL applications, it is becoming apparent that these types of explanations are not sufficient to explain the behavior of complex agents in many real-world settings (Puiutta & Veith, 2020; Madumal et al., 2020). For example, the inherent time-dependent characteristic of RL's decision making process can not be easily captured by feature-based methods. In the second line of work, XRL techniques help the user to understand the agent's dynamic behavior (Yau et al., 2020; Cruz et al., 2021; Juozapaitis et al., 2019). The main underlying idea of this class of XRL methods is to reveal to the user how the agent 'views the future' as most algorithms compute an explanation using various forms of the agent's future beliefs like future rewards, goals, etc. For example, (Juozapaitis et al., 2019) proposed to modify a DQN agent to decompose its Q-function into interpretable components. (van der Waa et al., 2018) introduce the concept of explaining two actions by explaining the differences between their future consequences.

In this work, we present CrystalBox, a novel framework for extracting post-hoc black-box explanations. CrystalBox is designed to work with input-driven RL environments which is a rich class of RL environments, including systems or networking domains. Input-driven environments have two distinctive characteristics compared to standard RL settings. These environments operate over input data traces (where a trace can be a sequence of network conditions measurements), and often have a decomposable reward function. Traces are difficult to model, and make both policy learning and explainability more challenging: learning a self-explainable policy can lead to significant performance degradation. Hence, we build CrystalBox on the principle of separation between policy learning and explanation computation. Our next key observation is that thanks to the decomposable reward property, we can adapt the idea of decomposable returns (Anderson et al., 2019) as the basis for explanations. Below, we summarize our main contributions.

1. We propose the first post-hoc black box explanation framework for input-driven RL environments.

2. We demonstrate that decomposable return-based explanations (Anderson et al., 2019) are a good fit for input-driven RL environments and propose a novel method for generating decomposed future returns using on-policy Q-function.

3. We design two complementary approaches to compute on-policy Q-function approximations outside of the RL agent's training loop: sampling- and learning-based methods.

4. We implement CrystalBox and evaluate it on input-driven RL environments. We demonstrate that CrystalBox produces high-fidelity explanations in real-world settings.

## 2 SYSTEMS ENVIRONMENTS

Systems environments are a rich class of environments that represent dynamics in computer systems, which are fundamentally different from traditional RL environments. We provide an overview of our representative examples, Adaptive Bitrate Streaming and Congestion Control, and various other systems environments. For a thorough discussion on these environments, please see Appendix A.2 In this section, we highlight the characteristics that we leverage in our explainer, decomposability of reward functions and the notion of traces in these settings.

**Adaptive Bitrate Streaming. (ABR)** In adaptive video streaming, there are two communicating entities: a client, such as a Netflix subscriber, who is streaming a video over the Internet from a server, such as a Netflix server. In video streaming, the video is typically divided into small seconds-long chunks and encoded, in advance, at various discrete bitrates. The goal of the ABR controller is to maximize the Quality of Experience (QoE) of the client by choosing the most suitable bitrate for the next video chunk based on the network conditions. The controller ensures that the client receives an uninterrupted high-quality video stream while minimizing abrupt changes in the video quality and stalling. QoE in this setting is typically defined as a linear combination that awards higher quality and penalizes both quality changes and stalling (Mok et al., 2011).

Note that network conditions are non-deterministic and constitute the main source of uncertainty in this setting. For example, the time taken to send a chunk depends on the network throughput. These network conditions are defined as the trace in ABR. More concretely, a trace is a sequence of network throughput values over time in ABR. Thus, an environment in ABR is modeled using network traces that represent network conditions.

**Congestion Control (CC)** Congestion control protocols running on end-user devices are responsible for adaptively determining the most suitable transmission rate for data transfers over a shared, dynamic network. When a user transmits data at a rate that the network cannot support, the user experience high queuing delays and packet losses. Deep RL-based solutions have shown superior performance in this setting (Jay et al., 2019; Abbasloo et al., 2020). Similar to the ABR environment, traces in this setting also constitute a timeseries of throughput values. The reward function in congestion control incentivizes higher sending rates and penalizes delay and loss.

**Other Systems Environments.** Deep RL offers high performance in cluster scheduling (Mao et al., 2019b), network planning (Zhu et al., 2021), network traffic engineering (Chen et al., 2018), database query optimization (Marcus et al., 2019), and several other systems control problems. A

common theme across these deep RL-based systems controllers is the decomposable reward function. The reason for that is that control in systems settings involves optimization across multiple objectives which are typically represented as the various reward components.

## 3 FORMALIZATION OF EXPLANATIONS

**Preliminaries**. In systems environments, we consider an Input-Driven Markov Decision Process (Mao et al., 2018), which is a special class of Markov Decision Processes where the environment transitions depend on an outside process called a *trace*. Formally, an Input-Driven MDP is defined by the tuple $(S, A, Z, P_s, P_z, r, \gamma)$, where $S$ is the set of states, $A$ is the set of actions, $Z$ is the set of time-variant traces, $r$ is the reward function, and $\gamma$ is the discount. $P_s(s_{t+1}|s_t, a_t, z_t)$ is the transition function of the environment that outputs the distribution of the next state, given the current state $s_t$, the action $a_t$, and the value of the current trace $z_t$. Finally, $P_z(z_{t+1}|z_t)$ is the transition function of the traces, which outputs the distribution over the next value of the trace given the current one. For example, in ABR, $P_z$ is a model which determines how the Internet link between the viewer and the platform behaves over time.

**Explainablity.** We take the perspective of systems operators. It is important to gain an understanding of a controller's decision-making process. Some of the common questions may be 'Why does the controller pick action A?', 'Why is action A better than action B?', and 'What are the measurable consequences of picking an action A?'. Note that these questions span from explanations about a single action to explanations that require reasoning about multiple actions. To answer these questions, we need to define a structure of explanations that is (a) succinct and (b) expressive.

Decomposed future returns (Anderson et al., 2019) is a category of explanations that satisfies these requirements. When each return component is meaningful to the user and the number of return components is small, decomposed future returns provide a concise and expressive explanation. This technique was demonstrated to be effective in learning a self-explainable agent in game environments (Juozapaitis et al., 2019). In systems environments, since the reward functions are naturally decomposable, the future returns are decomposable as well. Moreover, each component represents an aspect of performance or cost that is meaningful to the operator. Thus, decomposed future returns are an apt choice as the units of explanation in this setting. The core challenge then is to generate these decomposed future returns accurately and efficiently.

**Future returns**. To build our explanations we require an oracle to compute future returns of a given state $s_t$, an action $a_t$, and a policy $\pi$. We note that this problem is equivalent to computing the decomposed on-policy $Q^\pi(s_t, a_t)$ function, which calculates the expected future returns for taking acting action $a_t$ in state $s_t$ and following the policy $\pi$ thereafter.

We propose to directly approximate this decomposed on-policy version of the Q-function, $Q^\pi$, outside of the policy training process. This separation allows us to build post-hoc explanations for any fixed policy $\pi$, even if $\pi$ is non-deterministic or has continuous action space. We only require to be able to query this policy, without ever having to modify or know its internal structure.

Following (Juozapaitis et al., 2019), we define the explainability problem as estimating the decomposed components of the on-policy action-value function $Q^\pi(s_t, a_t) = \sum_{c \in C} Q_c^\pi(s_t, a_t)$, where $C$ is the set of reward components. For example, in ABR, the components are Quality, Quality Change, and Stalling. Each component $Q_c^\pi(s_t, a_t)$ computes the expected return of that component for taking action $a_t$ in state $s_t$ and following policy $\pi$ thereafter. It is formally defined as:

$$Q_c^\pi(s_t, a_t) = r_c(s_t, a_t) + \mathbb{E}_{s_{t+1}, a_{t+1}, \ldots \sim \pi} \sum_{\Delta t=1}^{\infty} [\gamma^{\Delta t} r_c(s_{t+\Delta t}, a_{t+\Delta t})], \forall c \in C \qquad (1)$$

We can obtain empirical samples of this function for all of the different components $c$ by Monte Carlo rollouts. We refer to these Monte Carlo samples of the ground truth as $\overline{Q_c^\pi}$ for convenience.

We define an explanation for a given state, action, and fixed policy as a tuple of return components:

$$\mathcal{X}(\pi, s_t, a_t) = [Q_{c_1}^\pi, \ldots, Q_{c_k}^\pi], \quad c_1, \ldots, c_k \in C \qquad (2)$$

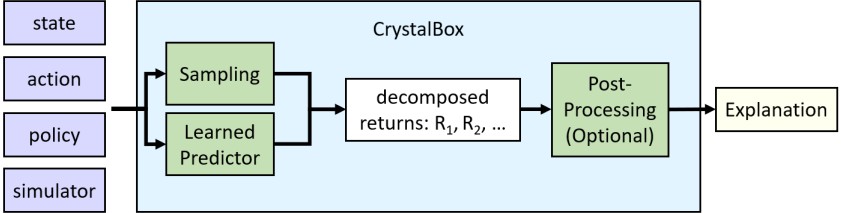

Figure 1: **Overview of CrystalBox**.

In general, one can consider more complex explanations that are functions over the return components. The function may depend on concrete environments and user preferences.

## 4 CRYSTALBOX

In this section, we present our novel framework, CrystalBox. We start with a high-level description. CrystalBox consists of two main components (Figure 1). The first component is the future returns predictor. It takes as inputs a state, an action, a simulation environment, and a policy. We present two ways to build this component, a sampling-based approach (§ 4.1–4.1.2) and a learning-based approach (§ 4.2). The predictor produces expected returns that are fed to an optional post-processing module which generates easy-to-understand explanations. As an example, we present a post-processing approach to summarize the returns in Section 5.3.

We discuss a few assumptions we make about available data. The framework requires four inputs: a state, an action, a policy, and a simulation environment. The first two inputs, a state, and an action, form a pair that we want to explain. The next input, policy, is treated as a *black-box that we can only query*. We never assume access to the model of the environment or future information such as $s_{t+1}$ or $z_t$. The only assumption we make is that we have access to a simulation environment, the last input. Note that for most input-driven RL environments, these simulation environments are publicly available, e.g., ABR (Mao et al., 2017), CC (Jay et al., 2019), network scheduling (Mao et al., 2019a).

Let us highlight several features of CrystalBox. First, it does implement our design principle which is the separation of policy learning and explanation computations. Second, it is flexible and allows the user to plug and play different environments and policies. To the best of our knowledge, CrystalBox is the first framework that provides such capabilities among the class of reward-based explainers. In the next section, we present a few approaches to design future returns predictors.

### 4.1 SAMPLING-BASED APPROACH

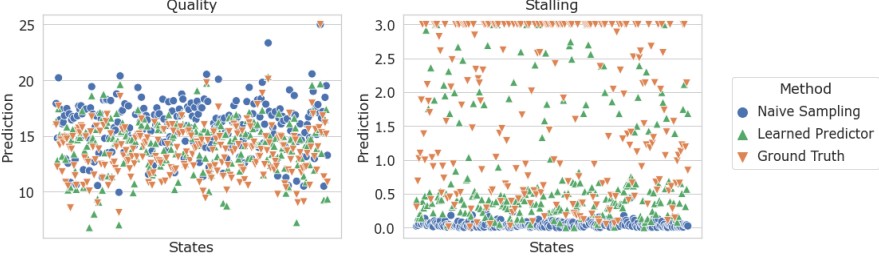

Figure 2: **Examples of future return predictions**: Examples of Quality and Stalling return predictions by Naive Sampling and Learned approaches, plotted alongside samples of ground truth returns. We see that Naive sampling fails to capture stalls or quality drops.

Our first approach to designing a predictor is sampling-based: we estimate the individual components of $Q^\pi(s_t, a_t)$ empirically by averaging over the outcomes of running simulations starting from $s_t$ and taking the action $a_t$. However, in practice, $Q_c^\pi(s_t, a_t)$ cannot be computed exactly. Thus, to get an empirical approximation of $Q_c^\pi$, it is necessary to bound the infinite horizon by a fixed length

Figure 3: **Overview of Learning-Based Approach**.

$t_{max}$. Enforcing this bounded horizon approximates the true $Q_c^\pi$ with a commonly used truncated version where the rewards after $t_{max}$ are effectively assumed to be zero (Sutton & Barto, 2018).

To approximate $Q_c^\pi(s_t, a_t)$, we need to sample potential futures of state $s_t$ for $t_{max}$ steps. If we have a model of $P_z$ available, we may simply use it to obtain samples of $z_t$, and in turn $s_t$. In this case, our sampling would be equivalent to the MC simulation method. However, in input-driven environments, $P_z$ is not available (Mao et al., 2018; 2019a). Therefore, to obtain potential futures of state $s_t$ we have no choice but to sample traces from $Z$. Evidently, it is not MC simulation anymore, as these potential futures are 'guessed' by our sampling procedure rather than given to us. $Z$ can be sampled using different strategies and we discuss two possible strategies.

Before diving into sampling strategies, we consider how sampling-based approach would work on ABR. Suppose we need an explanation for a drop in bitrate in ABR. In this case, we roll out the policy $\pi$ in the environment and consider a set of states with a drop in bitrate for the next chunk. Our goal is to approximate $Q_c^\pi(s_t, a_t)$ in these states using our sampling strategies. Note that we can also continue rollouts from this point onwards and compute $\overline{Q}_c^\pi$ (§ 3) of each of the future return components. We can use these to gain an initial understanding of how good our approximations are.

### 4.1.1 NAIVE SAMPLING

A simple strategy for sampling involves uniformly random sampling. Given a state $s_t$, we randomly sample traces from $Z$ to guess potential futures and compute approximations of $Q_c^\pi$. Now we can compare computed approximations of $Q_c^\pi(s_t, a_t)$ values with $\overline{Q}_c^\pi$. Figure 2 shows results of the comparison for the ABR example above focusing on two return components: quality and stalling. We observe that naive sampling-based predictions have low accuracy, especially for stalling predictions. To analyze the poor performance of the naive sampling approach, we took a close look at the sampling procedure. Recall that we randomly sample $Z$ to obtain potential futures, so our estimates depend on the distribution of $Z$. We observe that the distribution of traces is very unbalanced (see Fig. 8 in Appendix A.1). The dominant traces do not sufficiently represent all relevant scenarios. One remedy to solve this issue is to make our sampling produce distribution aware, e.g. we could weight potential futures that we get from $Z$.

### 4.1.2 DISTRIBUTION-AWARE SAMPLING APPROACH

We propose an improved sampling-based method. As we mentioned in the previous section, a sampling-based approach can benefit from a smarter weighting of potential futures that we obtain from $Z$. To do so, we take advantage of the features of $s_t$ and condition our future values by calculating $P(z_t|s_t)$. In practice, this probability distribution cannot be easily computed because of the complexity of the underlying environment. We propose a method to approximate this conditioning. We assume that traces have underlying natural clustering, e.g. clusters may correspond to a set of regions, clients, time, etc. Hence, we cluster all traces in $Z$ and provide a procedure to map the state $s_t$ to its closest cluster. Finally, we randomly sample a trace within that cluster. In an experimental evaluation, we demonstrate that such conditioning does improve the sampling-based method.

### 4.2 LEARNING-BASED APPROACH

Our second approach is learning-based. This approach is based on the insight that future returns components of $Q^\pi(s_t, a_t)$ form a function that can be directly parameterized and learned in a model-free manner by a function approximator.

The proposed learning procedure consists of two phases (see Figure 3). In the first phase, we take a policy and a simulation environment and collect trajectories by rolling out policy $\pi$ in the simulation environment. Next, we pre-process the trajectories to create a dataset of $(s_t, a_t, Q_c^\pi(s_t, a_t))$ tuples. In the second phase, we learn our predictor $Q_{c,\theta}^\pi$ for each component, where $\theta$ is a set of neural network parameters. We emphasize that we employ deep *supervised* learning to find the final parameters $\theta$ by iteratively updating the function approximator to better approximate the samples of $Q_c^\pi$. $Q_{c,\theta}^\pi(s_t, a_t) \leftarrow Q_{c,\theta}^\pi(s_t, a_t) + \alpha(Q_c^\pi(s_t, a_t) - Q_{c,\theta}^\pi(s_t, a_t))$. Here, $Q_{c,\theta}^\pi(s_t, a_t)$ is the prediction of the neural network, and $Q_c^\pi(s_t, a_t)$ is a sample of the true value of the Q-function, calculated in the first phase by looking at the trajectory of states and actions after $s_t$ and $a_t$. As in the sampling-based approach, we use the truncated version of the $Q_c^\pi$ function.

This formulation is a special case of the function approximation version of the Monte Carlo Policy Evaluation algorithm (Silver, 2015; Sutton & Barto, 2018) for estimating $Q_\theta^\pi$. In our case, $Q_\theta^\pi$ is further broken down into smaller return components $Q_{\theta,c}^\pi$ that can be added up to the original value. Therefore, the standard proof of correctness of the Monte Carlo Policy Evaluation applies. Thus, our method converges to the true $Q^\pi$ function and captures how the policy performs.

### 4.3 QUALITY OF EXPLANATIONS

Next, we discuss evaluation metrics for explanations. First, we briefly overview commonly used evaluation criteria for explanations: the fidelity metric. In standard explainability workflow, an explainer takes as input a complex function $f(x)$ and produces an interpretable approximation $g(x)$ as output. For example, $g(x)$ can be a decision tree that explains a neural network $f(x)$. To measure the quality of the approximation, the fidelity metric $FD = \|f(x) - g(x)\|, x \in \mathcal{D}$ measures how closely the approximation follows the original function under an input region of interest $\mathcal{D}$.

Let us consider how these evaluation criteria are applied to our RL settings to evaluate CrystalBox explanations. It turned out that such a translation is rather direct. As above, we have the complex function $Q_c^\pi$, one per each component $c$ (defined in Section 3.). CrystalBox outputs it approximation, i.e. a predictor $\mathrm{Pred}(Q_c^\pi)$, that also serves as an explanation. Hence, the fidelity metric is defined as a norm between a complex function and its approximation:

$$FD_c = \|Q_c^\pi - \mathrm{Pred}(Q_c^\pi)\|, \forall c \in C. \tag{3}$$

In our experiments, we use $L_2$ norm. However, there is one distinction to discuss. Unlike standard settings, $Q_c^\pi$ is neither explicitly given to us as input nor can be efficiently extracted in any realistic environment, e.g. systems environments described in Section 2. Hence, the best we can do is to obtain estimates of $Q_c^\pi$ using Monte Carlo rollouts.

## 5 EVALUATION

We present an experimental evaluation of CrystalBox that consists of two parts. First, we evaluate the fidelity of the returns predictors described in Sections 4.1.1–4.2. Next, we focus on the explainability capabilities of CrystalBox.

We perform our experiments on two systems environments: ABR and CC. In ABR, the controller decides the video quality of an online stream to show to a client. The controller receives a reward equal to the quality of experience of the client, measured as a weighted sum of three components: quality, quality change, and stalling. In CC, the controller manages the Internet traffic of a connection between a sender and a receiver by adjusting the sending rate of the outgoing traffic from the sender. Here, the controller receives a reward that is a weighted sum of three components: throughput, latency, and loss. For a detailed overview of these environments, see § 2.

We consider the three return predictors we presented earlier: the naive-sampling based approach (§ 4.1.1), the distribution-aware sampling approach (§ 4.1.2), and the learning-based approach (§ 4.2). Moreover, in some applications, we may have partial access to the policy. For example, we may have access to the embeddings of the states. In this case, it is important for explainability frameworks to take advantage of this additional knowledge to improve explanations. We demonstrate that our approach can do this without any major modifications. Therefore, in addition to the "black-box" setting, we consider a "gray-box" variant of our learned predictor where our predictor reuses the embedding $\phi(s_t)$ of a state $s_t$ from the policy.

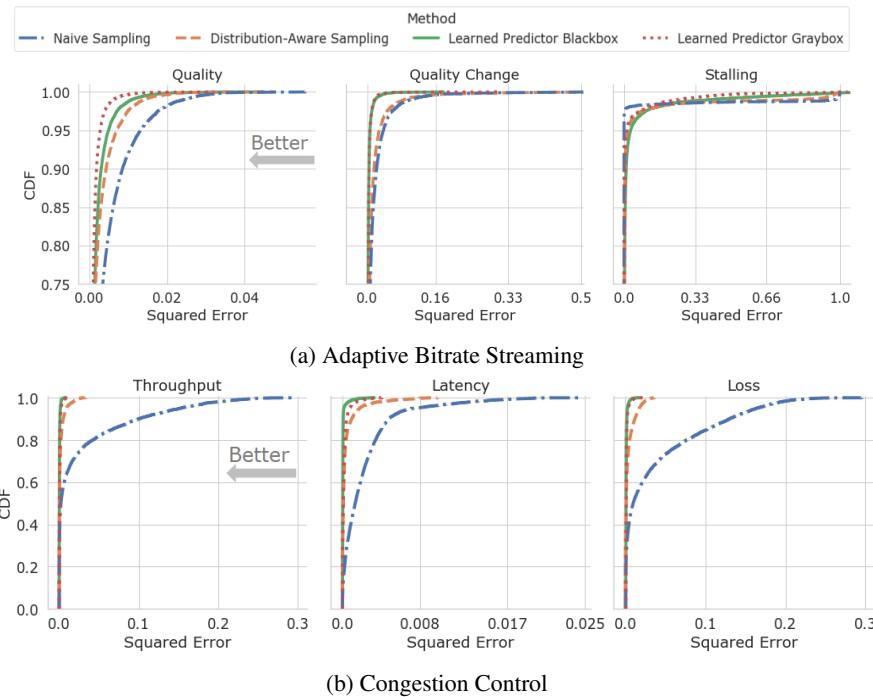

(a) Adaptive Bitrate Streaming

(b) Congestion Control

Figure 4: **Evaluation of CrystalBox for factual actions**: Distribution of Squared Error of different methods to Monte Carlo samples of the ground truth in Adaptive Bitrate and Congestion Control. Here, we focus on traces that can potentially experience stalling and discuss results on all traces in Figure 14 (Appendix A.6). The Learned approach offers predictions with the lowest error to the ground truth in all three return components of both environments. Note that the values of all the returns are scaled to be in the range zero to one before being measured for error. The y-axis in results for ABR is adjusted due to the inherent tail-ended nature of ABR's optimization.

## 5.1 FIDELITY EVALUATION

We recall that decomposable future returns form the basis for CrystalBox explanations, so it is critically important for us to produce accurate predictions. To measure the quality of these predictions, we turn to the fidelity metric we introduced earlier (§ 4.3), and measure the error between the predictions of different approaches and samples of the true $Q_c^\pi$ function. We generate these samples by rolling out the policy on a held-out set of traces $Z'$ to ensure that these samples have not been seen by any of the approaches before.

In Figure 4, we see that the learned predictor outperforms both of the sampling approaches in producing high-fidelity predictions of all three of the return components in both of the environments. The gray-box predictor narrowly beats the black-box approach at predicting the returns in Adaptive Bitrate Streaming, while achieving similar performance in Congestion Control. Next, we analyze the performance of two sampling-based methods. We see that Distribution-Aware sampling provides dramatic performance improvements over the standard sampling approach, especially, in CC. These results provide additional evidence to confirm our observation that exploiting the information in state $s_t$ can be vital to producing high-fidelity return predictions.

## 5.2 EXPLAINABILITY ANALYSIS

Having established the fidelity of our return predictors, we now turn to evaluating the explanatory power of these predictors. We focus our analysis on the predictors' ability to answer contrastive questions such as "Why action A instead of action B?". Recent work (Doshi-Velez et al., 2017; van der Waa et al., 2018; Mittelstadt et al., 2019; Miller, 2019) has highlighted the importance of such questions for human interpretability. Contrastive queries allow the user to differentiate between multiple possible actions to take, e.g. for debugging purposes.

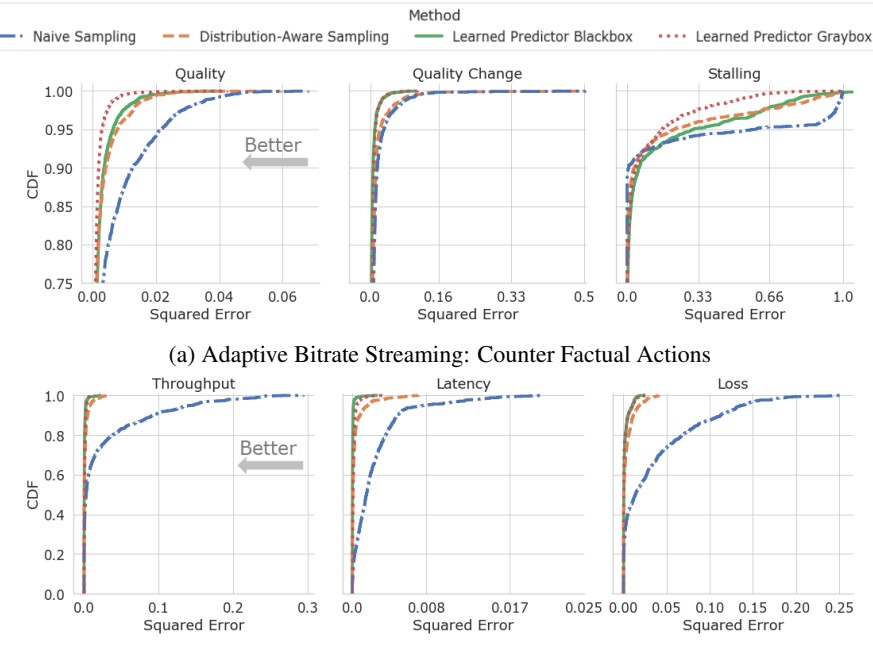

(a) Adaptive Bitrate Streaming: Counter Factual Actions

(b) Congestion Control: Counter Factual Actions

Figure 5: **Evaluation of CrystalBox for Counter-factual actions**: Distribution of Squared Error to samples of the ground truth decomposed return for ABR and CC. We see that the Learned approach offers the most accurate predictions for both factual and counter-factual actions in all of the different return components. Note that the values of all the returns are again scaled to be in the range zero to one before being measured for error. The y-axis in results for ABR is adjusted due to the inherent tail-ended nature of ABR's optimization.

CrystalBox supports answering this type of question. Given two actions A and B, we simply need to compute one explanation for A and one for B. The intuition is that by looking at explanations the domain expert can gain insight into why one action, e.g. the action that the policy suggests, is preferred over an alternative action. For example, consider ABR environment. Suppose that we are streaming a high-quality video. In state $s_1$, the policy unexpectedly drops to sending medium-quality video (action A). An alternative action B is to keep the same bitrate value which might be seen as a better action by the operator. To resolve this discrepancy, the user requests explanations for A and B from CrystalBox that we show in the following table:

| Action | Explanation (future returns per component) | | |
|--------|---------|----------------|----------|
| | quality | quality change | stalling |
| A | 16.23 | **0.85** | **0.0** |
| B | **16.31** | 2.11 | 0.41 |

Table 1: Example of predictions of returns

By comparing future returns, we see that action B is more preferable to action A in terms of video quality but it loses to A in terms of quality change and stalling ( we want these values to be as low as possible for QoE). These indications should convince the operator that action A is more reasonable in this situation. We would like to note that while our explanations operate in terms of very high-level notions for a given environment, like video quality for ABR, however, we do expect the systems operator to have the basic domain knowledge to draw conclusions given our explanations.

To quantitatively measure the quality of counterfactual explanations we again use the same fidelity metric. However, there is a difference between factual and counterfactual explanations that we need to take into consideration. Consider a set of trajectories generated in Phase 1 of the learning-based approach (see Figure 3). They come from running a given policy in the environment. Thus, only actions taken by the policy are recorded. However, we envision a range of use cases, like policy

debugging, where the user might be interested in actions that policy does not frequently take. In this case, such counterfactual actions might be underrepresented in these trajectories leading to poor future returns estimates. To resolve this issue, we again exploited our separation principle between policy training and learning a predictor. Namely, we augmented our dataset by generating additional trajectories where we add an explorative action to the beginning of the trajectory. We use the augmented dataset to train a single predictor. To clarify, the same predictor was used for factual (in Figure 4) and counterfactual explanations. In Figure 5 we observe the distribution of squared error of the different approaches to samples of the true $Q_c^\pi$ function where all the actions $a_t$ are counterfactual. We emphasize that counterfactual actions can be seen as difficult-to-predict scenarios for the reasons we just explained. First, we see that the learned predictor outperforms sampling-based approaches in almost all cases. Moreover, they provide *high-fidelity* return predictions for counterfactual actions. Another interesting conclusion is that we see the advantage of the gray-box over the black-box learned predictor in the same cases that were not that prominent for factual actions. Consider, for example, results for stalling in factuals in Figure 4a and counterfactuals in Figure 5a. The gray-box learned predictor significantly outperforms all other predictors in the latter plot.

### 5.3 EVENT DETECTION

CrystalBox provides an optional post-processing capability on top of original future return-based explanations. Consider again the example in Table 1. In many applications, return-based explanations can be sufficient. However, we believe that domain-specific post-processing can be very useful in practice. In some cases, it might not be obvious how to compare two contrastive actions in a state $s_1$ from just their numerical returns. For example, if the stalling return value under action B is 0.1 then it is unclear whether we should interpret it as a sufficient indication of stalling that is not present under action A. Rather than making the user wonder about how to compare these future returns, we can post-process them in a form of binary events, e.g. if a stalling happens or not in the near future.

We introduce the notion of *threshold* for demarcating the boundary between binary events along each return component. For example, in ABR environment, we use the 0.3 threshold for stalling. If the return value is greater than 0.3 then the explanation signals that a stalling occurs in the future. Thresholds can be determined based on a variety of factors such as risk tolerance, recovery cost for certain events, etc. In Appendix A.5, we show experimental evaluation of this technique, where we analyze two comparative actions: factual and counter factual from the same state. Overall, we show that all predictors are capable to detect a large portion of events, while the learning-based predictors have better recall of events.

## 6 DISCUSSION AND FUTURE WORK

We start our discussion with an applicability scope of CrystalBox and, then, discuss its possible extensions. In this work, CrystalBox targets systems-related applications. However, input-driven environments are not limited to this class of applications. For example, there is a rich class of game-based environments that are also input-driven (Mao et al., 2018). CrystalBox can be potentially extended to game-based environments, however, such extension is non-trivial. For example, in our learned approach, we used Monte Carlo returns as estimates of the ground-truth $Q_c^\pi$ function. However, in games where rewards can be extremely delayed (only at the end of the episode) or attributed to a large sequence of actions, these returns can be extremely high variance. Such high variance can lead to poor estimates of future returns, hence, low-fidelity explanations. To overcome this variance, several strategies can be explored (Mao et al., 2018; Hessel et al., 2018; Silver et al., 2017). We believe that it is an interesting future direction on its own.

One interesting direction to explore is whether we can use model distillation techniques to extract an interpretable model of future returns predictors. Another potential avenue is to explore whether we can employ future return predictors during policy learning to facilitate understanding and debugging for human-in-the-loop frameworks.

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

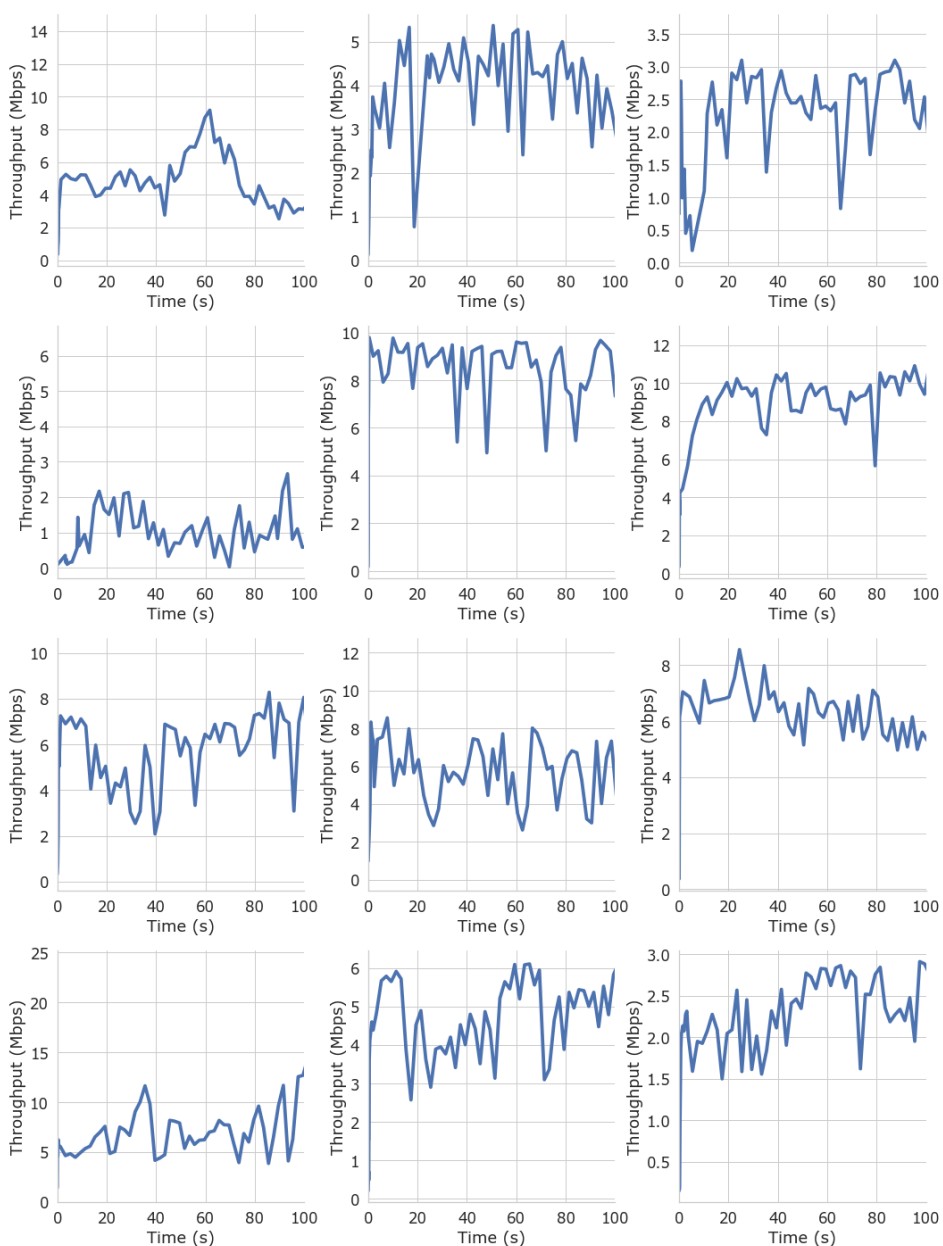

Figure 6: **Examples of Traces in Adaptive Bitrate Streaming**. In ABR, a trace is the over-time throughput of the internet connection between a viewer and a streaming platform. In this figure, we present a visualization of a few of those traces for the first 100 seconds. Note that the y-axis is different on each plot due to inherent differences between traces.

# A  APPENDIX

## A.1  TRACES

In this section, we visualize representative traces in Figure 6 and Figure 7 for ABR and CC applications, respectively.

In ABR, a trace is the over-time throughput of the internet connection between a viewer and a streaming platform. We obtain a representative set of traces by analyzing the logged data of a public live-streaming platform (Yan et al., 2020). In Figure 6 we present a visualization of a few of

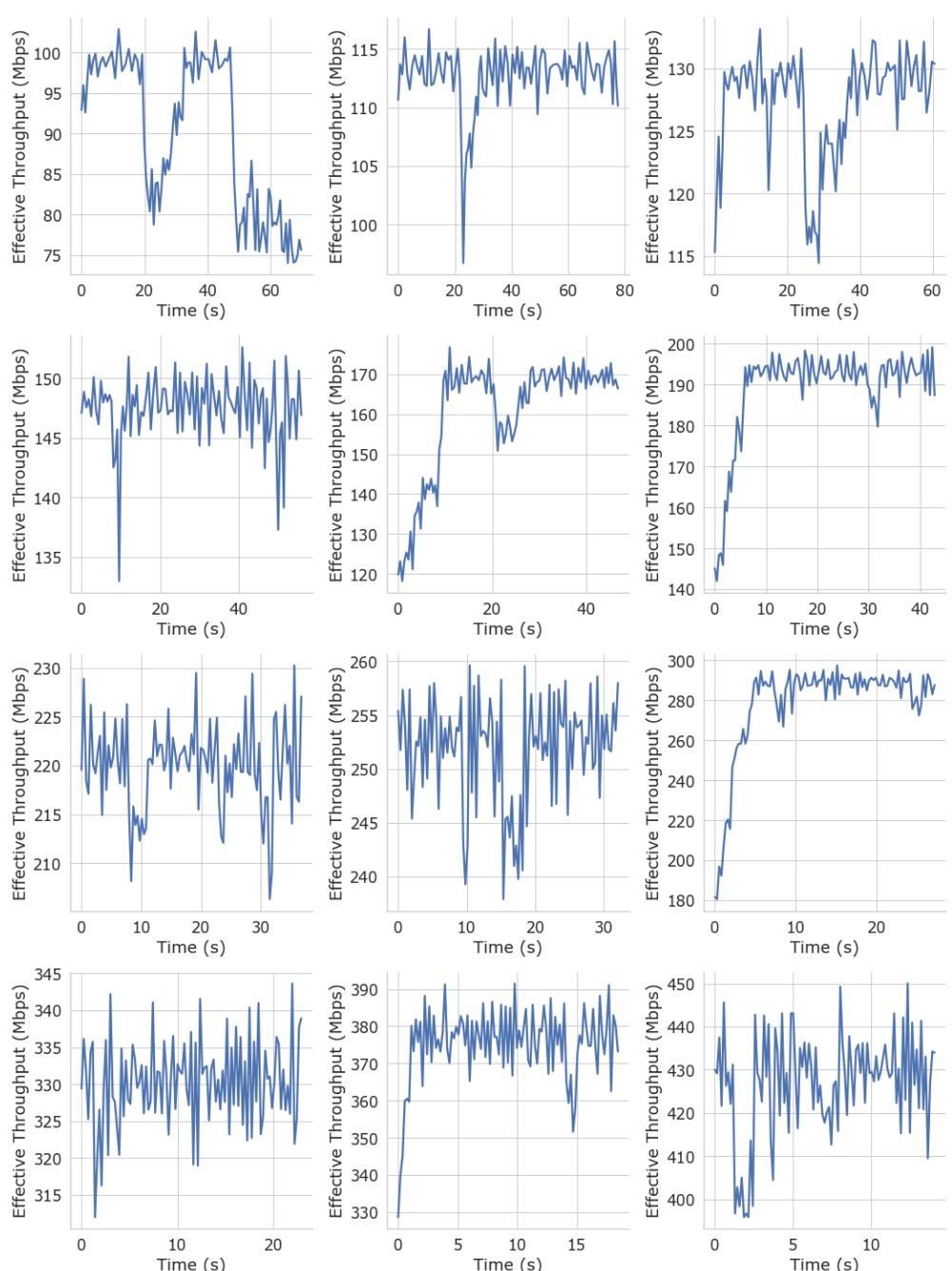

Figure 7: **Examples of Traces in Congestion Control**. In CC, a trace is defined as the internet network conditions between a sender and a receiver over time. These conditions can be characterized by many different metrics such as throughput, latency, or loss. In this figure, we represent these traces by the sender's effective throughput over time. Note that both the x-axis and y-axis are different on each plot due to the inherent differences between traces.

those traces for the first 100 seconds. Note that the y-axis is different on each plot due to inherent differences between traces. However, even with a naked eye, we can see that some traces are high-throughput traces, e.g. all traces in the third row, while other traces are slow-throughput, e.g. the first plot in the second row. To further analyze these inherent differences, we analyze the distribution of mean and coefficient of variance of throughput within each trace. In Figure 8, we see that a majority of traces have mean throughput well above the requirements of the highest quality video. When we

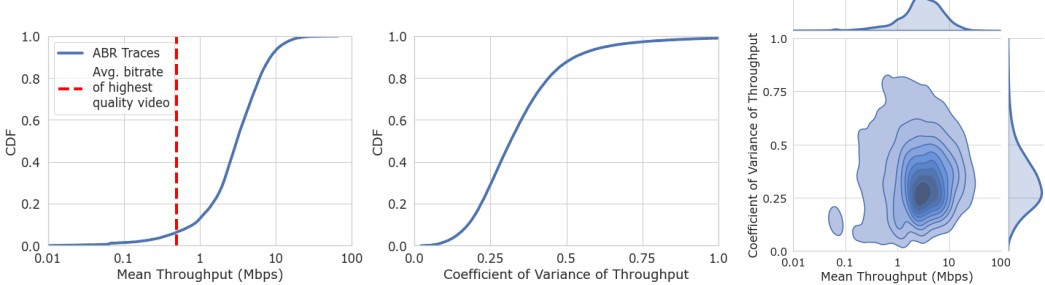

Figure 8: Distribution of Traces in ABR. **Left**: distribution of the mean throughput in traces. Note that the x-axis is log-scaled due to the large differences between all the clients of this server. **Middle**: distribution of coefficient of variance of the throughput within each trace. **Right**: The joint distribution of mean and coefficient of variance of throughput. The traces are logged over the course of a couple of months from an online public live-streaming Puffer (Yan et al., 2020). We find that a majority of the traces have mean throughput well above the bitrate of the highest quality video. Only a small percentage of traces represent poor network conditions such as low throughput, high variance, etc.

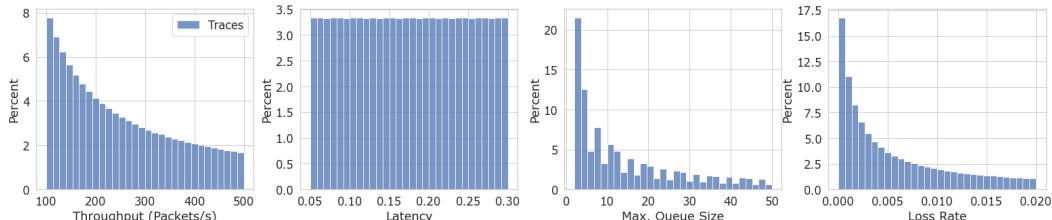

Figure 9: **Distribution of Traces in CC**: We analyze the distribution of traces in CC by analyzing the distribution by four key metrics: throughput, latency, maximum queue size and loss. These traces are synthetically generated by sampling from a range of values, similar to the technique employed by (Jay et al., 2019). We observe that traces with especially poor network conditions such as high loss rate or high queuing delay are small in number.

analyze this jointly with the distribution of throughput coefficient of variance, we see that a majority of those traces also have smaller variance. Only a small number of traces represent poor network conditions such as low throughput or high throughput variance. These observations are consistent with a recent Google study (Langley et al., 2017) that showed that more than 93% of YouTube streams *never* come to a stall.

In CC, a trace is the over-time network conditions between a sender and receiver. We obtain a representative set of traces by following (Jay et al., 2019) and synthetically generating them by four key values: mean throughput, latency, queue size, and loss rate. In Figure 7, we demonstrate how these traces may look like from the sender's perspective by looking at the effective throughput over time. Similar to the traces in ABR, we can visually see that the traces can be greatly different from one another. In Figure 9, we analyze the effective distribution of these traces. We observe that while the distribution isn't nearly as unbalanced as it is in ABR, there are still only a small number of traces that have exceedingly harsh network conditions.

## A.2 Environments in the Systems Domains

In system domains, traditional environment simulators are not available due to the complexity of the underlying input-process $z$. To circumvent this issue, the state-of-the-art training environments replay specific logged runs of the system called "traces". The training environment selects a specific input-trace $z$ from the set of given traces, and generates the successive state $s_t$ by simply looking up the logged value $z_t$ to compute $P(s_{t+1}|s_t, a_t, z_t)$. This replaying process circumvents the need to explicitly calculate or approximate the behavior of the trace $P(z_t|z_{t-1})$. However, this replaying process is not available when the policy is deployed in the real world. In the real world, we do not know the future value of the input-trace $z_t$, nor do we have a way to approximate it using $z_{t-1}$.

To summarize, we highlight the key differences between traditional simulators and "simulators" in input-driven environments.

- **Rollout mechanism during training:** In traditional simulators, future states can be sampled. In trace-based simulators, it is not obvious how to obtain future states in a similar manner because they depend on an underlying process for which we lack a model. To overcome this issue, the state-of-the-art solution is to select a specific trace and replay it. This is a reasonable substitute for having a model of the underlying process and is efficient.
- **Complexity of rollouts:** Rollouts can be expensive in traditional simulators. However, in trace-based simulators, given a specific trace, trace replay in the environment is relatively inexpensive, we just replay this trace.
- **Rollout mechanism during evaluation:** In traditional simulators, this mechanism is not different from the one during training. Therefore, Monte Carlo sampling of the future states is trivial. By contrast, in trace-based environments, the rollout mechanism from point 1 cannot be used, because we are not given the future nor do we have a method to model it.

## A.3 Examples of future return predictions (additional figures)

We visualize all three components of future returns in Figure 10 which is an extension of Figure 2.

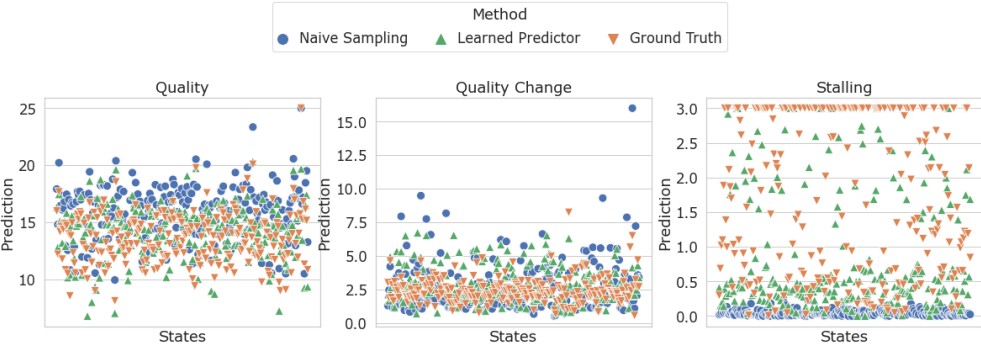

Figure 10: **Examples of future return predictions**: We visualize the Quality Change component of the future return predictions we presented in Fig. 2. We present Quality and Stalling here again for clarity. Unlike the other two return components where Naive sampling achieves dramatically poor performance, we see that Naive Sampling can detect quality changes to a certain degree.

## A.4 Runtime Analysis

We visualize the latency of Sampling-based and Learning-based techniques to generate an explanation. In sampling-based techniques, we empirically estimate the return components $Q_c^\pi$ by rolling out the policy for $t_{max}$ steps under a number of sampled traces $z_1, z_2, z_3, \ldots$. During this rollout process, we repeatedly query the simulator and policy at each step. By contrast, in Learning-based techniques, we directly generate the return components by querying a neural network once. In Figure 11, we see the impact of this reduced overhead: learning-based techniques offer drastically lower latency than sampling-based techniques, reducing the computation time from 50-250 ms to just 5-6 ms.

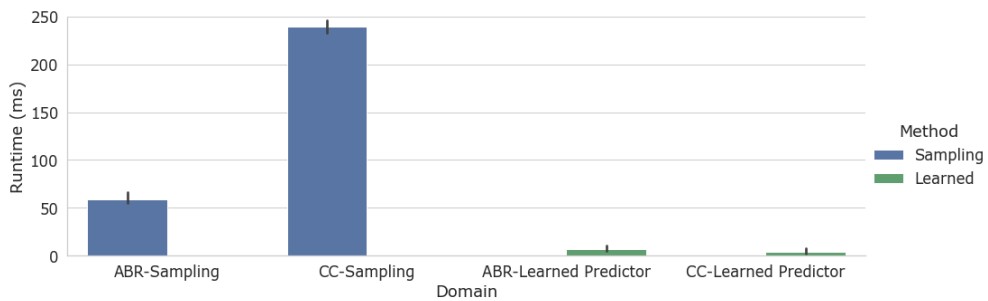

Figure 11: **Runtime Analysis**: We visualize the amount of time taken by Sampling-based and Learning-based techniques to generate an explanation in both ABR and CC environments. We see that the Learning-based techniques significantly reduce the compute time necessary to generate an explanation, reducing it by an order of magnitude on average.

## A.5   EVENT DETECTION

We present our evaluation of an event-based post-processing technique. We recall that the goal of the post-processing technique is to make an explanation easier for the operator to interpret. To do so, we introduce a notion of a threshold to convert numerical returns to event-based returns. For example, if the value of future return for quality change is greater than 0.2, we say that an event of quality change is going to be seen in the near future.

To evaluate the effectiveness of this post-processing technique, we consider the scenario where the operator may want to understand the impact of two comparative actions in the same state. We select a trace $z$, rollout the policy in that trace, and select a state $s_t$ to focus on. Within that state, we simulate taking two actions, $a_{t_1}$ and $a_{t_2}$, and continue rolling out the policy using $z$ under both the actions to obtain two $\overline{Q}_c^\pi$ vectors.

Our event-based post-processing allows us to treat the explainer as an event predictor model. Hence, we can evaluate it using standard metrics. The ground truth events are obtained from the two $\overline{Q}_c^\pi$ vectors using the same threshold. Note that threshold values are parameters of this post-processing method. We manually tried out a few threshold values and choose one that balances recall and false positives among those we tried on the training set.

Figure 12 shows our results for ABR (the first row) and CC (the second row). The first plot in each row shows results for factual and the second for counterfactual explanations. For ABR we used the following event threshold values: {quality below 0.55, quality change above 0.2, stalling above 0.3}. For CC we used the following event threshold values:{throughput below 0.4, latency above 0.15, loss above 0.025}. On the y-axis, we show the percentage of events that were correctly detected by each explanation generation method. For example, if there are 10 quality drop events and naive sampling detects 5 of them then the percentage value is 50 as this method detects 50% of events. Consider ABR results first. For factual explanations, sampling-based methods are better in detecting large quality change events. However, learning-based methods are better in detecting quality drop and long stall events. In fact, sampling-based method misses all long stall events. If we consider results for counterfactuals we see that learning-based methods outperform sampling methods. We observe a slightly different picture in CC environment. While the learned-based approaches outperform all other at detecting high latency, all the predictors provide high performance in other scenarios.

Next, we compute the false positive rate for each predictor. Figure 13 shows these results for ABR and CC environments. By false-positive we mean a situation when a predictor signals that an event will happen but this is not the case w.r.t. $\overline{Q}^\pi$. As can be seen from the plot, the false-positive rate is low for all predictors overall. The only exception that stands out is the high loss events in CC. Note that learning-based methods mostly manage to avoid false positives on factual explanations in CC. However, they do have a high false positive rate on counterfactuals.

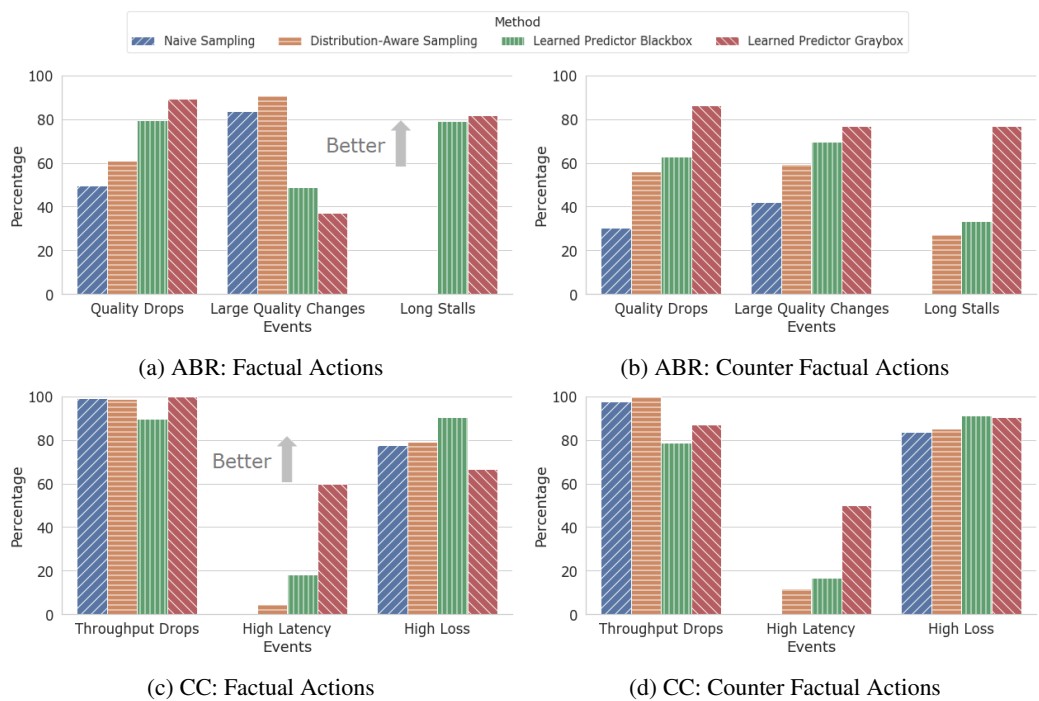

Figure 12: **Event Detection**: We analyze the efficacy of different predictors at detecting events. We identify events happening by detecting if samples of the ground-truth return exceed a threshold. For a detailed discussion, see Section 5.3. We evaluate their efficacy by analyzing their recall under both factual and counterfactual actions.

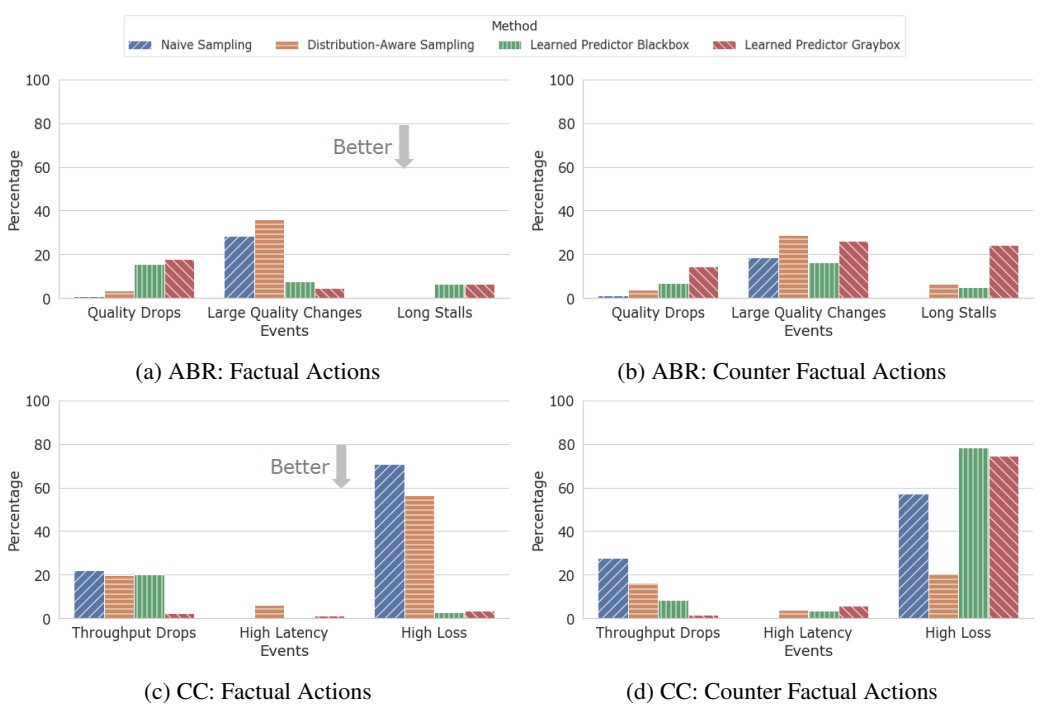

Figure 13: **False Positive Rates on Event Detections**: The false positive rate of the different predictors under both factual and counterfactual actions. We consider predictions false-positive when ground-truth samples of returns do not exceed the threshold, but the predictions exceed it.

### A.6 FIDELITY EVALUATION (ADDITIONAL RESULTS FOR ABR)

We present our evaluation of CrystalBox explanations on all traces. Figure 14 shows our results. We can see that all predictors perform really well. The learned-based predictors do slightly outperform sampling-based but the difference is not that prominent compared to results on traces that might experience stalling (see Figure 4a). For high throughput traces, the optimal policy for the controller is simple: send the highest bitrate. Therefore, all predictors do well on these traces.

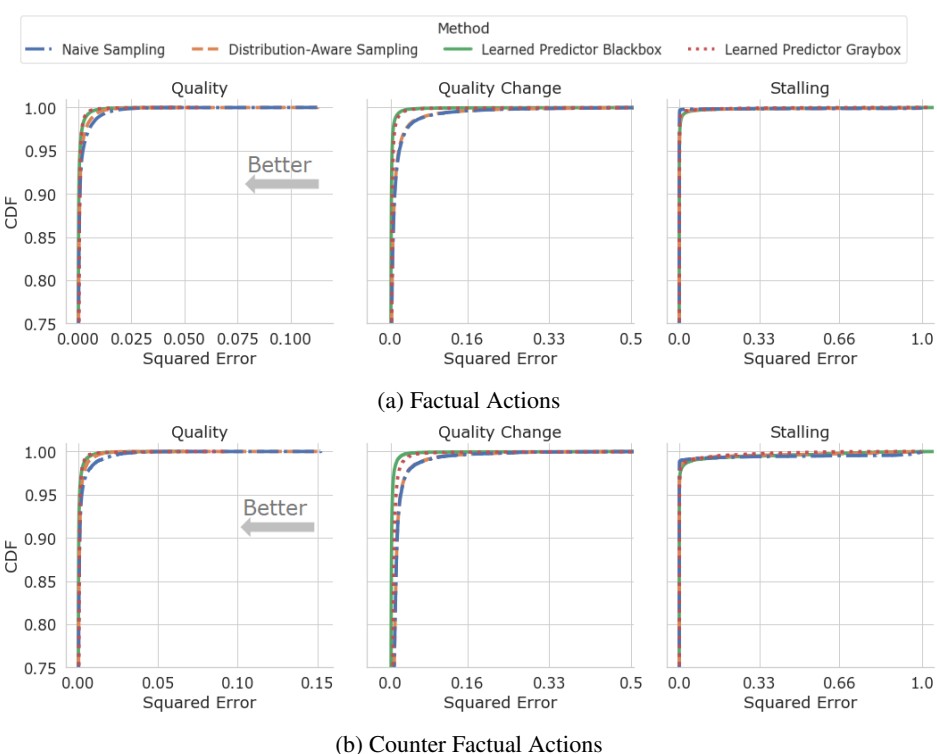

Figure 14: **Evaluation of CrystalBox in ABR across all traces**. Distribution of Squared Error to samples of the ground truth decomposed return predictions for all traces in ABR. We observe that the differences in distribution of error for all of the return predictors shrink, but the relative ordering remains the same. Learned approach offers the best predictions for both factual and counterfactual actions.

### A.7 MONTE CARLO ROLLOUTS

We collect samples of the ground truth values of the decomposed future returns by rolling out the policy in a simulation environment. That is, we let the policy interact with the environment under an offline set of traces $Z$, and observe sequences of the tuple $(s, a, \vec{r})$. With these tuples, we can calculate the decomposed return $Q_c^\pi(s_t, a_t)$ for each timestamp. However, for a given episode, these states and returns can be highly correlated (Mnih et al., 2013). Thus, to efficiently cover a wide variety of scenarios, we do not consider the states and returns $Q_c^\pi(s_t, a_t)$ after $s_t$ for $t_{max}$ steps. Moreover, when attempting to collect samples for a counterfactual action $a_t'$, we ensure the rewards and actions from timestamp $t$ onwards are not used in calculation for any state-action pair before $(s_t, a_t')$. This strategy avoids adding any additional noise to samples of $Q^\pi$ due to exploratory actions.

$t_{max}$ is a hyper-parameter for each environment. In systems environments, we usually observe an effect of each action within a short time horizon. For example, if a controller drops bitrate, then the user experiences lower quality video in one step. Therefore, it is only required to consider rollouts of a few steps to capture consequences of each action, so $t_{max}$ equals to five is sufficient for our environments.

### A.8 LEARNING-BASED APPROACH

**Preprocessing**. We employ Monte Carlo Rollouts to get samples of $Q_c^\pi$ for training our learned predictor. By themselves, the return components can vary across multiple orders of magnitude. Thus, similar to the standard reward clipping (Mnih et al., 2013) and return normalization (Pohlen et al., 2018) techniques widely employed in Q-learning, we normalize all the returns to be in the range [0, 1].

**Neural Architecture Design**. We design the neural architecture of our learned predictors to be compact and sample efficient. We employ shared layers that feed into separate fully connected 'tails' that then predict the return components. We model the samples of $Q_c^\pi$ as samples from a Gaussian distribution, and predict the parameters (mean and standard deviation) to this distribution in each tail. To learn to predict these parameters, we minimize the negative log likelihood loss of each sample of $Q_c^\pi$.

For the fully connected layers, we perform limited tuning to choose the units of these layers from {64, 128, 256, 512}. We found that a smaller number of units is enough in both of our environments. We present a visualization of our architectures in Figures 15-18.

**Learning Parameters**. We learn our predictors in two stages. In the first stage, we train our network end-to-end. In the second stage, we freeze the shared weights in our network, and fine-tune our predictors with a smaller learning rate. We use an Adagrad optimizer, and experimented with learning rates from 1e-6 to 1e-4, with decay from 1e-10 to 1e-9. We tried batch sizes from {50, 64, 128, 256, 512}. We found that small batch sizes, learning rates and decay work best.

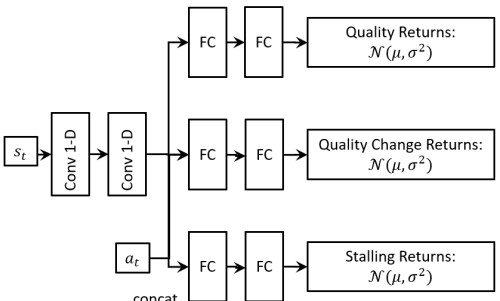

Figure 15: **Neural Architecture of the Black-box Learned Predictor in ABR**.

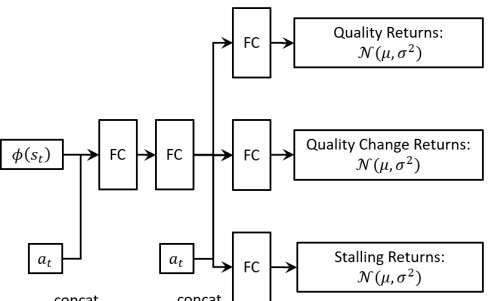

Figure 16: **Neural Architecture of the Gray-box Learned Predictor in ABR**.

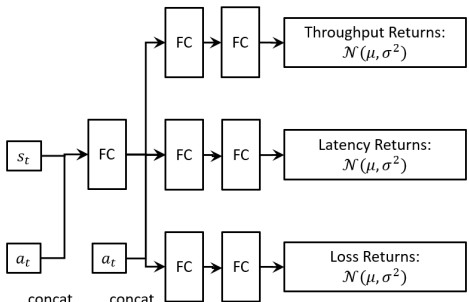

Figure 17: **Neural Architecture of the Black-box Learned Predictor in CC**.

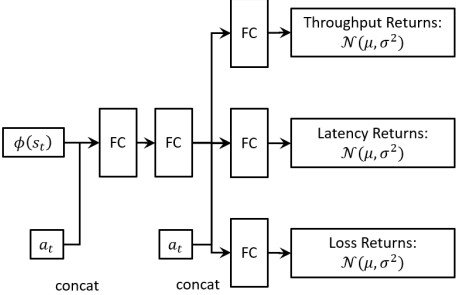

Figure 18: **Neural Architecture of the Gray-box Learned Predictor in CC**.

