# OpenReview forum: "CrystalBox: Efficient Model-Agnostic Explanations for Deep RL Controllers"
_ICLR.cc/2023/Conference — Submitted to ICLR 2023_

### Official Review · Reviewer_NU4N · 2022-10-21

**Confidence:** 4
**Correctness:** 4
**Technical Novelty And Significance:** 2
**Empirical Novelty And Significance:** 2
**Recommendation:** 5

**Clarity, Quality, Novelty And Reproducibility:**

Clarity:
* The system is clearly described and the paper reads well.

Quality:
* The work is of high quality, but the experiments are lacking in evaluation of the system for true interpretability/explainability in the loop with humans. The proposed method is sensible, but requires access to a high-fidelity simulator for the given problem, which may be a severely limiting assumption.
* The failure cases of CrystalBox are not discussed or evaluated. For example, how does the explanation handle out-of-distribution states? When the policy degrades, can explanations highlight this degradation or will they deceive humans into over-trusting the policy [2].

Originality:
* Counterfactuals are not a new idea, and related work on counterfactuals has fewer limiting assumptions than CrystalBox. However, CrystalBox provides explanations (including counterfactuals) through reward decomposition, and the reward decomposition comes from high-accuracy reward predictors, offering an advantage over prior work when reward decomposition is available.

**Strength And Weaknesses:**

Strengths:
* The paper clearly defines the environment, problem setting, and explanations in the given context.
* The proposed method works well in the given settings with decomposable rewards, and discussions of extension to future work are readily addressed in the end of the paper.
* CrystalBox is evaluated for faithfulness to ground-truth reward values and achieves high correlation to ground-truth data. The system's explanations are sensible and feasibly useful.

Weaknesses:
* There is not an evaluation of how such an evaluation could be used with a human in the loop (e.g., [1, 2]). While fidelity/faithfulness is useful, it would be more meaningful to understand how these explanation are received by humans in relation to other explanations (e.g., counterfactuals without CrystalBox [3], or decision-tree approximations [4, 5] of the policy)
* The assumptions of CrystalBox are quite strict (e.g., requiring a simulator and decomposable reward function + labels for the Q function)
* There is no evaluation of runtime, though the method seems to potentially be quite slow (as there must be several simulated forward passes and reward-component predictions).
* CrystalBox fundamentally assumes that the policy is taking steps to maximize reward components in the given time-bound of the simulators rollouts, but this assumption may not always be valid (e.g., if the agent encounters out-of-distribution states and takes sub-optimal actions). For cases in which the policy may not be completely trustworthy, it seems that CrystalBox may also not be completely trustworthy.

[1] Silva, Andrew, Mariah Schrum, Erin Hedlund-Botti, Nakul Gopalan, and Matthew Gombolay. "Explainable Artificial Intelligence: Evaluating the Objective and Subjective Impacts of xAI on Human-Agent Interaction." International Journal of Human–Computer Interaction (2022): 1-15.

[2] Andrew Anderson, Jonathan Dodge, Amrita Sadarangani, Zoe Juozapaitis, Evan Newman, Jed Irvine, Souti Chattopadhyay, Matthew Olson, Alan Fern, and Margaret Burnett. 2020. Mental Models of Mere Mortals with Explanations of Reinforcement Learning. ACM Trans. Interact. Intell. Syst. 10, 2, Article 15 (June 2020), 37 pages. https://doi.org/10.1145/3366485

[3] Karimi, Amir-Hossein, Gilles Barthe, Borja Balle, and Isabel Valera. "Model-agnostic counterfactual explanations for consequential decisions." In International Conference on Artificial Intelligence and Statistics, pp. 895-905. PMLR, 2020.

[4] Bastani, Osbert, Yewen Pu, and Armando Solar-Lezama. "Verifiable reinforcement learning via policy extraction." Advances in neural information processing systems 31 (2018).

[5] Wu, Mike, Michael Hughes, Sonali Parbhoo, Maurizio Zazzi, Volker Roth, and Finale Doshi-Velez. "Beyond sparsity: Tree regularization of deep models for interpretability." In Proceedings of the AAAI conference on artificial intelligence, vol. 32, no. 1. 2018.

**Summary Of The Paper:**

The paper proposes a method for explaining black-box RL models called CrystalBox, developed primarily for Systems Environments (such as adaptive bitrate streaming and congestion control). CrystalBox involves using a simulator and trained policy to rollout different actions and predict reward-component values and produce explanations by presenting humans with decomposed return values for a given action selection. The proposed method is evaluated on fidelity/faithfulness to ground-truth reward component values, and an example explanation is shown to demonstrate how the system would work.

**Summary Of The Review:**

CrystalBox is an interesting idea and provides useful explanations, though the assumptions inherent to the method are difficult to overlook. The approach seems to be accurate in the domains in which it was evaluated, though there is no true evaluation with humans (who would be the actual end-users of explanations) and the failure cases of the explanations are not clearly explored. If the assumptions of having access to a simulator to generate counterfactuals could be relaxed, I would increase my score significantly, but at the moment the requirement of a simulator is too restrictive for the method to be more widely useful.

---

> ### Author Response · Authors · 2022-11-16
> **Response to initial review**
>
> Thank you for your time and constructive review. We address your concerns as follows:
>
> 1. **User Study on CrystalBox.** Please see point [4]@[reply-to-reviewer1](https://openreview.net/forum?id=K1CNgCJtLLr&noteId=wPI10AVOXl4) for a discussion on conducting a user study on CrystalBox.
>
> 2. **Strict Assumptions.** While it is true that CrystalBox requires a set of traces, a trace-driven simulator and a decomposable reward function, we do not assume access to ground-truth labels of Q^\pi function.  We obtain samples of the Q^\pi function by rolling out the policy in the simulator.
>
> 3. **Runtime Evaluation.** Thank you for pointing this out. We will add an analysis of the runtime of our sampling and learning techniques in Appendix A.4. To summarize our findings, sampling based techniques take anywhere from 50-250 ms to generate an explanation, while learning based techniques take only about 5-6 ms.
>
> 4. **Out-of-distribution states.**
> A state that is out-of-distribution to the policy is by extension also out-of-distribution to our explainer. We propose to overcome this limitation by exploiting the separation between policy learning and explaining, and adding noise to the actions taken by the policy when collecting the training data for our learned predictor.  This allows us to expose our explainer to limited  out-of-distribution states. In case of a significant distribution shift, the explainer needs to be retrained, and so may the policy.

---

### Official Review · Reviewer_24j8 · 2022-10-24

**Confidence:** 4
**Correctness:** 2
**Technical Novelty And Significance:** 2
**Empirical Novelty And Significance:** 2
**Recommendation:** 3

**Clarity, Quality, Novelty And Reproducibility:**

My impression is that the presented method is not limited to Input-Driven MDPs.
This is a positive in that this method is more applicable, but this also means that this method is (1) framed in an unnecessarily narrow way and (2) not as novel as initially suggested.
The need for P_z is seemingly resolved in 4.2, but "the trajectory of states and actions" is used for each estimate.
If these trajectories are available, what is the issue with P_z access in 4.1?
My understanding is that 4.2 only differs in using a neural network to generalize.
Once the P_z difficulty is resolved, state-action-state pairs are obtained as with a typical MDP; the difference from typical MDPs disappears.
From this point, how do the traces play into the explainability method?
What is the benefit of splitting out Z from S in this setting?
What are the advantages of this approach over other reward decomposition methods?

The evaluation should be improved. Using fidelity does not convey the information necessary.
Beyond that, using Monte Carlo rollouts for the ground-truth is odd. The presented experiment is comparing the error between two estimators that both should converge.
Why are the Monte Carlo rollouts not used directly as a method?
If a ground truth is needed, then consider making an environment where one can be computed.
If seeking to convey that a good estimate can be reached efficiently, then results for different numbers of samples should be presented.
Fidelity would be useful to measure if learning Qs during training (and comparing to Qs obtained after training that are derived from many more rollouts).

For Section 5.2, claims are made about using CrystalBox for answering comparative questions, but the evaluations use fidelity here, too.
To support the use of CrystalBox in comparing actions, pairwise evaluations (best action vs second-best, for example) are needed.

Other Comments:
- Figure 2 is not legible as a scatterplot. Also, a quantitative metric should be reported rather than simply visualizing the distributions.
- Figure 4 is oriented in an odd way (better being "further left").

**Strength And Weaknesses:**

This work uses reward decomposition, which is a promising subtype of XRL methods.
It highlights specific questions that the user seeks to address and returns to these questions in evaluation!

The presented approaches require a simulator and do not learn the decomposed reward function from replay buffer data nor during training.
The authors highlight the separation of learning and explaining, but:
1. This separation is not a desired property, as it removes the causal nature of explanations: the learned Q is no longer the reason for the policy behavior.
   With this method, one can get explanations where the sum of Q components would suggest an alternate action (relative to what agent performs) or an action ordering based on one of the reward components may differ from one that would be obtained during training.
   Note that an estimator will exhibit more error in rare states (whether rare in environment or ones where agent makes an unexpected choice).
   These situations are exactly the ones where an operator would like to see an explanation. As a result, this post-hoc approximation will disproportionately affect meaningful queries.
   This aspect of the method is not evaluated.
2. The transition to a post-hoc approximation of Q via traces is not difficult nor non-obvious.

This work focuses on input-driven environments, which are defined as using traces and typically have decomposable reward.
However, it is unclear why there are complexities with P_z. The method assumes that a simulator is available during one portion, but then P_z is said to be inaccessible at a different point.
Specifically, it is stated that the "framework requires four inputs: a state, an action, a policy, and a simulation environment ... [and we assume] we have access to a simulation environment, the last input."
It is unclear how the lack of a P_z simulator fits with this characterization of the required inputs and the assumptions made.
Why is Z directly used for traces rather than gathering samples during training?

A post-processing approach is presented in Section 5.3.
This is an important step in tailoring this approach to the specific problem type considered.
Developing these ideas further would strengthen this work.

**Summary Of The Paper:**

This work performs reward decomposition with a focus on Input-Driven MDPs.
It presents CrystalBox, which creates post-hoc explanations for blackbox RL agents in input-driven environments.
This method is evaluated on two tasks.

**Summary Of The Review:**

This work focuses on Input-Driven MDPs, but it ultimately presents a method that is not limited to Input-Driven MDPs.
As a result, this work is effectively performing reward decomposition except with approximators learned after-the-fact.
This aspect risks generating explanations that differ from the agent's own estimates.
Finally, this method is evaluated only in its ability to obtain estimates of future reward components.
The ability to use these decompositions in answering specific questions is not evaluated.

I have read the author responses. I appreciate the clarifications, but key concerns remain: my recommendation is unchanged.

---

> ### Author Response · Authors · 2022-11-16
> **Response to initial review**
>
> We thank the reviewer for the valuable comments. We address the main concerns as follows:
>
> 1. **Causal nature of explanations.** We emphasize that our settings are black-box, so we don't have access to either the training procedure or network parameters. We do believe that these settings are practically important for the adoption of RL controllers, as they are for classification tasks, where they are widely adopted. While such settings do have an inherent limitation of not capturing the casual nature of explanation in some cases, they can provide a broad understanding of the policy’s decision making process. There is a large body of work on blackbox explainability, e.g.  [a-c], demonstrating that this is an important research direction for a wide range of research communities, including systems.
>
> 2. **Access to P_z during evaluation and straight-forwardness of sampling.** Please see points [1] and [3] @[reply-to-all-reviewers](https://openreview.net/forum?id=K1CNgCJtLLr&noteId=en25hUl-Yv) for clarification on access and the problems in applying Monte Carlo sampling in a straight-forward manner.
>
> 3. **On the use of MC samples as ground truth.**
> To get ground-truth values of Q starting from a particular state, we must calculate the probability over the next states and rewards. However, in trace-driven environments, we cannot calculate this probability since it depends on the value of the trace z_t: we only have access to z_{t-1} and cannot model the underlying process of z to calculate the probability P(z_t | z_{t-1}). The best we can do is seed our trace-driven simulator with a particular logged trace z and rollout what would happen under that trace (please see point [1] @[reply-to-all-reviewers](https://openreview.net/forum?id=K1CNgCJtLLr&noteId=en25hUl-Yv)). In our evaluation, we replicate the fact that we do not have access to z_t by evaluating the explainers on a separate set of “testing” traces never seen before by any of the techniques.
>
> 4. **Evaluation of Comparative actions.** Thank you for pointing this out. We have analyzed the exact scenario that Reviewer 24j8 requested on pairwise evaluations. Namely, we compare the quality of different techniques at predicting the outcomes of 2 different actions given the same state in the Appendix A.3. Unfortunately, we did not fully describe the settings for this experiment in Section 5.3. We clarified this in the revised version of our paper.
>
> [a] G Liu, O Schulte, et al. Toward interpretable deep reinforcement learning with linear model u-
> trees. In ECML-KDD, 2018.
>
> [b]  Z. Meng, M. Wang, J. Bai, M. Xu, H. Mao, and H. Hu. Interpreting deep learning-based networking systems.  In Proc. ACM Special Interest Group Data Commun. Appl. Technol. Architect. Protocols Comput. Commun. (SIGCOMM), 2020, pp. 154–171.
>
> [c] Arnaud Dethise, Marco Canini, Srikanth Kandula:
> Cracking Open the Black Box: What Observations Can Tell Us About Reinforcement Learning Agents. NetAI@SIGCOMM 2019: 29-36

---

### Official Review · Reviewer_iPCV · 2022-10-27

**Confidence:** 4
**Correctness:** 3
**Technical Novelty And Significance:** 2
**Empirical Novelty And Significance:** 2
**Recommendation:** 3

**Clarity, Quality, Novelty And Reproducibility:**

This work is extremely clear and seems to be reproducible. The quality is fairly high in that the proposals are technically sound, though the experimental evaluation could be tightened. The novelty is on the lower side, as the core techniques and ideas have been explored previously, though not in conjunction.

**Strength And Weaknesses:**

## Strengths

This work's main strengths are that it studies an interesting and important problem of providing interpretability to RL policies, which are generally difficult to interpret, and that the presentation is quite clear.

## Weaknesses

- My main concern is that there is very limited technical contribution of interest to the ML community. Ultimately, the two proposed variants of CrystalBox leverage extremely well-understood classic techniques in fairly straightforward ways that can be found in RL textbooks, e.g., Sutton & Barto. I appreciate that these techniques are being used to decompose a value function into components, which is somewhat novel, but also still explored in prior work, e.g., Anderson et al., '19.
- The sampling-based method is also quite expensive and relies on fairly strong assumptions in general. Specifically, it requires many Monte Carlo roll-outs (which are expensive), and also requires the ability to reset to a given state, which is generally not possible across RL tasks, although it is true that in the considered input-driven tasks, there is access to a simulator that enables this. Yet, I assume that a downstream goal is to apply this in production systems, rather that merely on simulators, in which case, it would not be possible.
- Though this work claims that directly decomposing the rewards in the policy can "lead to significant performance degradation," it's not actually clear to me that this is the case. It seems quite possible that directly learning a separate value function for each reward component, or predicting different components as an auxiliary task could yield a better policy. At least, this requires some empirical investigation, and raises the question of why we need a separate framework to estimate these returns.
- Finally, a major concern is that it's difficult to assess the practical utility of the system from the current set of experiments. While the cdf error curves are interesting and show an ordering between the different proposed variants, it's unclear what amount of squared error is acceptable and useful for actually using this system for interpretability. Compounding this issue is the fact that even the computed squared errors are only estimates because the "ground-truth" values themselves are MC estimates. This work proposes reasonable ways that such a system might be used, but the key question remains: Is this system in fact useful for e.g., network engineers? A user study could potentially be useful here, although I do not have the expertise to understand how a network engineer might want to change the ABR or CC policy or its decisions based on the outputs of the interpretability system. However, determining how the interpretability system can beneficially impact future decision making is crucial -- if no decisions can practically be made from its outputs, then how is the interpretability helping?
- Minor comment: Section 4 states that CrystalBox consists of 2 components, but only describes 1 component.

**Summary Of The Paper:**

This work considers the problem of providing interpretability for RL agents in input-driven environments. Specifically, this work proposes CrystalBox, a method for decomposing expected future returns into multiple components, when the reward itself is composed of multiple components. This work presents two approaches for providing the decomposed expected future returns: 1) A sampling-based approach that estimates the future returns with Monte Carlo simulation, assuming access to a simulator that enables multiple roll-outs from the current state; 2) A learning-based approach, which implements Monte Carlo policy evaluation.

**Summary Of The Review:**

Overall, I am mainly concerned about the relevance and the impact of this work for the ML community. Therefore, I cannot recommend acceptance.

---

> ### Author Response · Authors · 2022-11-16
> **Response to initial review**
>
> We thank the reviewer for helpful insights and critique.
>
> Regarding the weaknesses mentioned:
>
> 1. **Limited contribution.**  We consider the explainability of RL agents in black-box settings. We believe it is a very important practical problem. First, there are no solutions in the literature that provide “future”-based explanations to the best of our knowledge.  Indeed, Andreson et. al. use a conceptually similar technique in white-box settings: decomposed reward-based explainability. We build on this work and make a significant step forward by proposing a solution that does not require altering the training procedure.
>
>
>     Second, sampling the future states in our setting *cannot be done* based on classical algorithms from Sutton & Barton. Unlike standard sampling solutions that use simulators to sample potential future states and rewards, we cannot sample future states because future states depend on the input value z_t (Please also see our clarification on the notion of a simulator@[reply-to-all-reviewers](https://openreview.net/forum?id=K1CNgCJtLLr&noteId=en25hUl-Yv)). However, current research techniques are unable to model P(z_t | z_{t-1}) due to the complexity of the underlying process. Therefore, our proposed technique to sample the future states is novel and a significant research contribution.
>
>
>     Third, our solution is general: we can work with both discrete and continuous action spaces. For example, Andreson et. al. can only work with discrete action spaces since it relies on a Deep Q-Learning agent. We make no assumptions about the RL algorithm or the environment setting.
>
>     We will highlight these contributions in the abstract.
>
>
> 2. **Strong assumptions about sampling.** Please see point [2]@[reply-to-all-reviewers](https://openreview.net/forum?id=K1CNgCJtLLr&noteId=en25hUl-Yv) for clarification on sampling efficiency.
>
>     Regarding access to a trace-based simulator for our explanation method (practical aspect). A large number of traces are publicly available so one can obtain a reasonable trace-based simulator.
>
> 3. **Claim on “significant performance degradation” for auxiliary decomposed reward prediction.**
> We mean that using decomposed reward during the training can make learning problems harder compared to single reward settings. We will downgrade this claim since we have not evaluated this scenario, as we focus on black-box explainability.
>
> 4. **Evaluation of our explainer, user study on network operators.**
> Evaluation of explainability techniques is a hard problem [a], there is no ground truth in many cases, no standard evaluation metrics, etc. The most common quantitative metric across many domains is the fidelity of the explanations, which we proposed to adapt to the RL settings. We believe that it is a fair evaluation of the explainer.
>
>     Regarding user study with network operators. While a thorough user-study with a large number of network operators can be a useful addition to our evaluation, we believe that it is out-of-scope of this work due to a number of reasons. Unlike many computer vision tasks, we cannot run such a study on Amazon Turk as we cannot find a subset of people with relevant expertise. Unfortunately, we don't have resources to run such a study in production settings. Moreover, recent studies show that user studies can often be misleading [b], and should not be used as a gold standard metric for any explainer.
>
>
> [a] https://explainml-tutorial.github.io/
>
> [b] Interpreting Interpretability: Understanding Data Scientists' Use of Interpretability Tools for Machine Learning, Kaur et al

---

> > ### Comment · Reviewer_iPCV · 2022-11-18
> > **Reviewer Response**
> >
> > I appreciate the authors' response and clarification. However, this work still appears to have severe weaknesses that prevent me from recommending acceptance.
> >
> > 1) Evaluation. I appreciate the authors' explanations for why a user study may be out of scope for this work. However, the broader concern of how to evaluate the system remains. Fidelity is a helpful metric, but presumably the whole point of an interpretability system is to help humans make better decisions based on the output of the system. It therefore seems crucial to provide some sort of signal about this. In the current form, it's entirely unclear to me how to make use of the systems' outputs, though, admittedly, I do not have network operator expertise.
> >
> > 2) Sampling assumptions. I appreciate that in trace-based simulators, sampling can be quite cheap. However, isn't the point to eventually run this on a real system, in which case the sampling will be extremely expensive again? Or is the idea that you always collect traces and train offline? I would appreciate further clarification on this point, though I acknowledge that today is the last day of the discussion period, and the authors may have limited time to respond.
> >
> > 3) Contribution. To further clarify, my understanding is that the base algorithms upon which this work builds are classic and known RL algorithms. However, specifically for the "distribution-aware sampling" approach, this work proposes a new way of sampling $z_t$ by clustering. Is this correct? I appreciate that this sampling strategy is novel, though I still have concerns about the contribution of this work, given that the core algorithms are known, and the new sampling strategy is very specific, and unlikely to apply to broader RL problems. This concern is also somewhat tied to the evaluation, as it's not entirely clear if this new sampling strategy provides significant benefits.

---

> > > ### Author Response · Authors · 2022-11-19
> > > **Response to Reviewer iPCV**
> > >
> > > We appreciate you thoroughly reading our response, and asking clarifying questions.
> > >
> > > 1. **Evaluation.**  Standard performance metrics developed by network engineers [a, b] are incorporated into the reward function, and act as building blocks in our explanations. Therefore, our explanations are natural and meaningful for network engineers. For example, when the controller takes a counter-intuitive action compared to human-designed heuristics, our explainer can help the network operator understand the behavior in terms of key performance metrics.
> > >
> > > 2. **Sampling Assumptions.** Yes, in both approaches, *we never perform online sampling* on real systems.
> > > (i) Learning-based explainers are trained offline, so sampling is not needed during deployment.
> > > (ii) Sampling-based approaches do sample from *a dataset of offline traces* during deployment.
> > >
> > > 3. **Contributions.** We present the first black-box explainer for RL controllers based on expected returns. We believe that it is a significant research contribution.  As the reviewer pointed out, our approach uses an adaptation of an existing RL algorithm to demonstrate our results. In the context of explainability, we see it as an advantage of our method as it can be easily incorporated into standard RL frameworks (we will release our code as well).
> > >
> > >
> > > [a] Yan, Francis Y., et al. "Pantheon: the training ground for Internet congestion-control research." 2018 USENIX Annual Technical Conference (USENIX ATC 18). 2018.
> > >
> > > [b] Yan, Francis Y., et al. "Learning in situ: a randomized experiment in video streaming." 17th USENIX Symposium on Networked Systems Design and Implementation (NSDI 20). 2020.

---

> > > > ### Comment · Reviewer_iPCV · 2022-12-08
> > > > **Response**
> > > >
> > > > I appreciate the further clarification and response. Overall, I do appreciate the contribution of providing a black-box explainer for RL controllers, where the reward naturally decomposes into several terms. However, I still think there are clear weaknesses. There is little technical contribution, since the methods are primarily grounded in well-known RL algorithms, though there is a new sampling component. Further, while I appreciate that the metrics used by network engineers are indeed incorporated into the algorithm, it is still not clear to me how such a system could be used to improve a networking system in practice. I think that this is a critical piece: there are many pieces of information we could gather about a system, but it seems important to make sure that the gathered information is actually practically useful. I would be in favor of accepting this paper if there were more significant evaluations that test this question, but in this current form I do not think the work is ready for publication.

---

### Author Response · Authors · 2022-11-16
**Response to all initial reviews**

Thank you to all the reviewers for your valuable comments and suggestions!

**Important clarification on 'simulators' in systems environments.**

In systems environments, traditional environment simulators are not available due to the complexity of the underlying process. To circumvent this issue, the state-of-the-art training environments replay specific logged runs of the system called “traces”. In systems literature, this method of simulating the environment is referred to as a “simulator” as well. This created confusion between traditional simulators and trace-based simulators.

We would like to highlight the main differences between traditional and trace-based simulators as they play a crucial role in our design decisions.
1. [Rollout mechanism during *training*] In traditional simulators, future states can be sampled. In trace-based simulators, it is not obvious how to obtain future states in a similar manner because they depend on an underlying process for which we lack a model. To overcome this issue, the state-of-the-art solution is to select a specific trace and replay it. This is a reasonable substitute for having a model of the underlying process and is efficient.
2. [Complexity of rollouts] Rollouts can be expensive in traditional simulators as Reviewer iPCV pointed out. However, in trace-based simulators, given a specific trace, trace replay in the environment is relatively inexpensive, we just replay this trace.
3. [Rollout mechanism during *evaluation*] In traditional simulators, this mechanism is not different from the one during training.Therefore, MC sampling of the future states is trivial, for example, as Reviewer 24j8 pointed out.  By contrast, in trace-based environments, the rollout mechanism from point 1 cannot be used, because  we are not given the future nor do we have a method to model it.

We will clarify the notion of a simulator in Appendix A.2.

---

### Decision · Program_Chairs · 2023-01-20

**Decision:**

Reject

**Justification For Why Not Higher Score:**

The reviewers raise valid concerns that have not been fully addressed in the rebuttal. I consider the lack of human evaluation as a major limitation of the present work.

**Justification For Why Not Lower Score:**

N/A

**Metareview: Summary, Strengths And Weaknesses:**

The paper addresses the problem of explainability of reinforcement learning based system controllers. This is motivated by the need for explanations in real-world applications of RL controllers. The proposed approach, CrystalBox, learns a separate explanation given a policy. The approach is evaluated in trace-based simulated real world settings of adaptive bitrate streaming (ABR) and congestion control (CC).

Initial reviews highlight the interesting, important and difficult nature of the problem addressed here. Giving useful explanations to users of such a system could genuinely advance real-world RL applications. In addition, an approach based on reward decomposition is considered promising.

A range of concerns and questions were raised in initial reviews, some of which were addressed during the rebuttal period. Novelty is considered limited, as the approach uses well-known techniques. Developing novel insights for new uses of well-known techniques could be a valuable contribution, but would require greater depth of insight than developed here. Reviewers are not convinced by the motivation for training an explanation model separately from the policy, especially given that such an explanation model could contradict the system's policy. Evaluation is not sufficient. Reviewers note that the provided metrics provide a valuable starting point, but the authors do not systematically show the value of generated explanations to human users. The discussion during the rebuttal phase did clarify questions about the nature of the trace-based simulator setup that is used here, and that is common in systems research.

Some of the raised concerns were not conclusively addressed during the rebuttal, with evaluation being a major example here. A revised version of the paper would need to systematically show how explanations generated by the proposed approach would provide value to users of such a system. Given the remaining concerns, the consensus is not to recommend acceptance at this stage.